# Development and Application of Real-Time PCR-Based Screening for Identification of Omicron SARS-CoV-2 Variant Sublineages

**DOI:** 10.3390/genes14061218

**Published:** 2023-06-02

**Authors:** Anna Esman, Dmitry Dubodelov, Kamil Khafizov, Ivan Kotov, German Roev, Anna Golubeva, Gasan Gasanov, Marina Korabelnikova, Askar Turashev, Evgeniy Cherkashin, Konstantin Mironov, Anna Cherkashina, Vasily Akimkin

**Affiliations:** 1Central Research Institute of Epidemiology, Federal Service for Surveillance on Consumer Rights Protection and Human Wellbeing Russian Federation, 3a Novogireevskaya Str., 111123 Moscow, Russia; 2Moscow Institute of Physics and Technology, National Research University, 115184 Dolgoprudny, Russia

**Keywords:** SARS-CoV-2 variants, Omicron, RT-PCR, surveillance

## Abstract

The Omicron strain is currently the main dominant variant of SARS-CoV-2, with a large number of sublineages. In this article, we present our experience in tracing it in Russia using molecular diagnostic methods. For this purpose, different approaches were used; for example, we developed multiprimer panels for RT-PCR and Sanger and NGS sequencing methods. For the centralized collection and analysis of samples, the VGARus database was developed, which currently includes more than 300,000 viral sequences.

## 1. Introduction

The number of substrains of the SARS-CoV-2 Omicron (B.1.1.529) variant, first identified in November 2021 in South Africa, is growing rapidly: more than 320 of them have already been identified worldwide (GISAID data), and more than 250 have been found in Russia. As of November 2022, the predominant sublineage in the territory of the Russian Federation, as in many other countries, is BA.5, although its proportion gradually began to decline, yielding to recombinant variants XBB* (includes descendent lineages).

The Omicron subseries contain a complex mutation profile that confers better affinity for the ACE2 receptor and increased resistance to various antibodies formed in response to vaccination or previous disease. 

The emergence of new sublineages and recombinant Omicron variants, such as XD, XE, XF, XL, XN, XP, XQ, XU, and XV, indicates that the process of virus evolution is actively going on. As long as no strain significantly different from Omicron displaces it, as happened when the Omicron strain replaced Delta, we can expect to see new Omicron subvariants in the future [1].

As compared to other SARS-CoV-2 variants of concern (VOC), the Omicron variant and its sublineages exhibit increased transmissibility and immune escape from neutralizing antibodies generated by previous infection or vaccination and cause numerous re-current and breakthrough infections [2]. 

Figure 1 shows the phylogenetic relationships of SARS-CoV-2 clades. Initially, Omicron included three main isolated sublineages (BA.1, BA.2, and BA.3), which were discovered almost simultaneously. Two more broad sublineages, BA.4 and BA.5, were later identified, in addition to many new sublineages within BA.1 and BA.2.

BA.4 and BA.5 are descended from BA.2, and their mutation profiles in the S protein gene are most closely associated with this variant. In addition to the mutations in BA.2, BA.4 and BA.5 have extra mutations, delHV69-70, L452R, F486V, and a wild-type amino acid at the position Q493 [3]. BA.4 and BA.5 have a similar mutational pattern in the 5′ region of the genome (from ORF1ab to Envelope) but show divergence in the 3′ region (from M to the 3′ end of the genome) [4]. It is suggested that BA.4 and BA.5 may have diverged by recombination, with a break between the E and M genes [3]. 

Importantly, despite the sequence identity of the S-protein gene, differences in other genome regions seem to confer a transmissive advantage on the BA.5 variant, which actually displaced BA.4 worldwide.

**Figure 1 genes-14-01218-f001:**
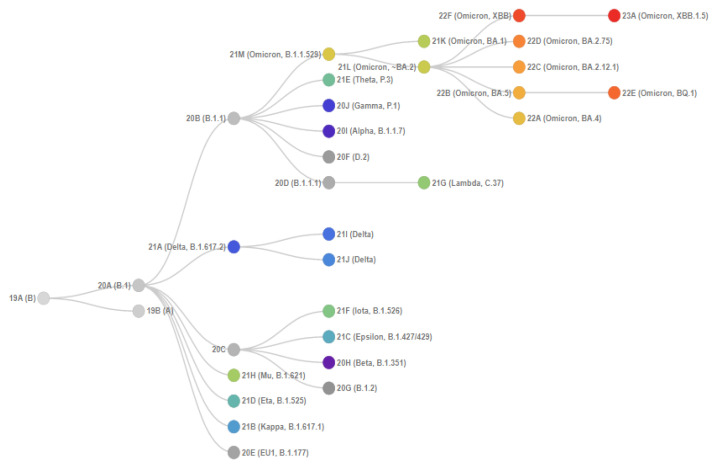
Phylogenetic relationships of Nextstrain SARS-CoV-2 clades. This figure is from CoVariants [5]. This project is free and open source [6].

Changes in the nucleotide sequence of the SARS-CoV-2 genome can be identified in various ways. Next-generation sequencing technologies are the primary tool for discovering and tracking SARS-CoV-2 variants, allowing whole-genome sequencing to provide the most comprehensive information [7].

A platform was created for depositing the obtained nucleotide sequences: the VGARus (Virus Genome Aggregator of Russia) database contains information about SARS-CoV-2 sequences and their common mutations for various regions of Russia and can be used to store, organize, and retrieve data for mutation detection and the identification of virus strains [8]. The overwhelming amount of genomic data was obtained in 2021 and especially in 2022, with a wide spread of the Omicron variant. Importantly, the volume of information in the VGARus database significantly exceeds the number of virus genome sequences from Russia in the GISAID database [9]. As of December 2022, VGARus contains more than 125,000 whole-genome sequences from Russia, in comparison to GISAID, which has about 60,000. At the same time, most of the whole-genome sequences of the CRIE (Central Research Institute of Epidemiology) were also deposited in the GISAID database and are freely available for download. 

A valuable feature of the database is its emphasis on epidemiology: nucleotide sequences are accompanied by non-personalized information about the epidemiological history, including information about the date and place of sample collection, sex and age of patient, etc. At the whole country level, the VGARus makes it possible to control the completeness of the coverage by the study of the population of each subject and optimize survey plans based on information about the time spent at each stage of work.

However, the next-generation sequencing approach can be quite laborious, it requires expensive equipment and reagents, it may take a long time, and new methods for faster and cheaper genotyping are also being developed to overcome these limitations [10]. At the same time, the strategy of using fragment sequencing (e.g., Sanger sequencing) or screening methods for virus subtype classification has also been successful in some instances [11,12].

The coordinated work of specialists from different fields, taking into account the results of whole-genome and Sanger sequencing and the use of screening in tracking variants, provides a complete epidemiological profile of the circulating virus. In this article, we present our experience in using molecular screening methods to rapidly track different SARS-CoV-2 Omicron variant sublineages and describe the specifics of their development and production.

Epidemiological analysis using VGARus allowed the development of screening techniques to differentiate Omicron sublineages.

We developed a multiplex assay for screening and detecting Omicron mutations. These mutations allow the differentiation of Omicron sublineages, such as BA.1 (delHV69-70, T95I, Ins214EPE, E484A, N501Y), BA.2 (delLPP24-26, E484A, N501Y), BA.3 (delHV69-70, T95I, E484A, N501Y), BA.4/BA.5 (delLPP24-26, delHV69-70, L452R, E484A, N501Y), and the Delta variant (L452R, P681R).

## 2. Materials and Methods

### 2.1. Biological Material Samples

Only anonymized samples of biological material, which were remnants from SARS-CoV-2 RNA testing, were used to develop the Omicron typing screening method.

Biological samples—swabs of the nasopharynx and oropharynx—were examined for the presence of SARS-CoV-2 RNA using the AmpliSens^®^ COVID-19-FL reagent kit (Amplisens, CRIE, Moscow, Russia). The study included samples with Ct values between 18 and 35 (which corresponds to virus concentrations of about 10^3^–10^4^ copies/mL). 

RNA isolation was carried out by nucleic acid precipitation using the “RIBO-prep” kit (Amplisens, CRIE, Moscow, Russia) according to the instructions for the kit.

Reverse transcription was performed using the REVERTA-L reagent kit (Amplisens, CRIE, Moscow, Russia) in accordance with the instructions of the manufacturer and Revertase (MMlv), RT-G-mix-2, Taq-F polymerase, and PCR-mix-2 FEP/FRT (all these reagents were from Amplisens, CRIE, Moscow, Russia) were used for one-step RT-PCR.

Standards methods were followed for working with biological samples [13]. Sanger sequencing and whole-genome sequencing were chosen as the methods of comparison for the developed methodology.

### 2.2. Whole-Genome and Sanger Sequencing Conditions

We used a modified ARTIC v.3.0 panel for whole-genome sequencing, which was modified to account for new genomic changes, as well as a panel of primers for sequencing the S protein gene only, which was described earlier [14]. Commercial solutions, such as the ATOPlex RNA Library Prep Set from MGI and Midnight kit from ONT (Oxford Nanopore Technologies, Oxford, UK), were used. 

Sanger sequencing was performed by the Sanger method on a 3500× Genetic Analyzer automatic sequencer (Applied Biosystems, Waltham, MA, USA) using reagents recommended by the manufacturer. We used an in-house primer panel for the amplification of SARS-CoV-2 S-gene fragments [15]. The characteristic amino acid substitutions of variants were determined with the sequencing of several fragments of the S-protein gene from 21,574 to 22,115 bp (5–185 aa) and from 22,790 to 23,302 bp (410–580 aa).

In the course of this work, we used protocols for sequencing complete genomes or the S-protein gene, which are described in our previous publications [14,16]. In general, all protocols involve the PCR amplification of parts of the virus genomes followed by sequencing on platforms from Illumina, Thermo Fisher Scientific, Oxford Nanopore Technologies, and MGI. Pangolin [17], proprietary tools, and in-house scripts were used for variant classifications.

The analysis of the obtained sequences was carried out either using the in-house developed software GEM:CoV-2 (the program for analyzing and processing nucleotide sequence fragments to detect mutations in the SARS-CoV-2 genome) or using the VGARus platform tools [18]. GEM:CoV-2 software analyzes the sequence chromatograms and creates the list of mutations compared to the Wuhan reference sequence MN908947. Two different approaches for virus genotyping are implemented in the VGARus platform: for fully sequenced genomes of SARS-CoV-2, the Pangolin tool [18,19] is used to determine a Pango lineage [20] and for sequences of S-protein gene, the in-house script is used. This script determines all mutations in the gene, comparing it with the Wuhan reference and thus choosing the appropriate lineage, using a table of typical mutations for the different lineages.

### 2.3. Data Analysis

The VGARus platform at the CRIE was used to assess the epidemiological prevalence of Omicron sublineages in Russia [18].

Figure 2 presents the scheme of interaction between different departments and the algorithm of the work performed. Thanks to the software tools integrated into the portal, organizations that do not have their own bioinformatic resources can obtain a detailed analysis of the uploaded genomes.

### 2.4. Real-Time PCR Target Selection

The following S-gene changes were selected for sublineage identification by real-time PCR in this study: delLPP24-26, delHV69-70, T95I, Ins214EPE, L452R, E484A, N501Y, and P681R.

For nucleotide sequence alignment for the design of primer–probe positions, BioEdit [22,23] was used. Omicron lineages’ whole-genome sequence data were obtained from the Global Initiative on Sharing Avian Influenza Data (GISAID) database [24]. Outbreak.info genomic reports [25] were used for lineage comparison and to estimate the mutation prevalence across lineages.

Two primers and an LNA-modified fluorescent probe were used in the reaction for each cDNA target to be amplified (RNA in the case of the multiplex assay). 

For single assays, the following protocol was used: 1 cycle of 95 °C for 15 min, followed by 45 cycles of 95 °C for 10 s/60 °C for 20 s. Fluorescence detection was performed at the 60 °C stage through channels for fluorophores FAM, R6G, and ROX.

The detection protocol for single PCR used the developed technique as follows: the reaction mix for each mutation detection consisted of a 10 µL PCR-mix-1 (contains dNTPs, primers, and probes; the concentration of primers and probes was experimentally selected from 0.5 to 0.7 µM and from 0.2 to 0.3 µM, correspondingly), 5 µL of PCR-mix-2 (contains Tris-buffer and MgCl2), and 0.5 µL of TaqF polymerase. Finally, 10 μL of cDNA sample or plasmid DNA positive control sample was added to the prepared tubes.

For one-step multiplex assays, the following protocol was used: holding 50 °C for 20 m, followed by 1 cycle of 95 °C for 15 min, followed by 45 cycles of 95 °C for 10 s/60 °C for 20 s. Fluorescence detection was performed at the 60 °C stage through channels for fluorophores FAM, R6G, and ROX.

The detection protocol for one-step multiplex assays used the developed methodology as follows: the reaction mixture for each mutation detection consisted of 10 μL of PCR-mix-1 (contains dNTPs, primers, and probes; the concentration of primers and probes was experimentally selected from 0.5 to 0.7 µM and from 0.2 to 0.3 µM, correspondingly), 5 µL of PCR-mix-2 (contains Tris-buffer and MgCl2), 0.5 µL of TaqF polymerase, 0.25 µL of Revertase-MMLV, and 0.25 μL of RT-G-Mix-2 (contains 1-Thioglycerol). Finally, 10 μL of RNA sample or plasmid DNA positive control sample was added to the prepared tubes.

The positive controls were obtained by cloning specific amplicons with target mutations in the pGEM-T vector (Promega, Madison, WI, USA) according to the manufacturer’s protocols. The correctness of the cloned sequences was confirmed by Sanger sequencing. Each fragment contained a detected mutation and flanking regions, which were used as primer locations. 

For the multiplex assay, control samples were mixed in equimolar quantities to obtain a single mixture of all the control samples used in the protocol.

The concentration of the single positive control samples was measured spectrophotometrically using the Nanodrop 2000. 

Single control samples with a concentration of 20 ng/μL were mixed in equal proportions. For the final mixture, a dilution series was prepared and analyzed by RT-PCR with all primer–probe mixtures used in a monoplex format.

According to the results of the experiments, the dilution that yielded Ct values in the range of 20–25 in PCR was chosen. Subsequently, all experiments were carried out with this dilution.

### 2.5. Development of Multiplex Reverse Transcription-Polymerase Chain Reaction (RT-PCR) for Genotyping of Omicron SARS-CoV-2 Sublineages

The development of real-time multiplex RT-PCR for screening genotyping of Omicron SARS-CoV-2 sublineages proceeded in several steps. 

The choice of targets (SARS-CoV-2 mutations) for detection was based on the prevalence of mutations in different Omicron sublineages. When selecting targets, we considered the significance of mutations, taking into account the evolution of the virus and their properties, as well as the difficulty of implementing detection within the framework of the approach used. The next steps are oligonucleotide synthesis, the testing of the systems in a single assay, the optimization of the conditions, and the testing of the system in a multiplex assay. 

To develop a screening technique for typing Omicron SARS-CoV-2 sublineages, S-gene regions containing mutations characteristic of sublineages BA.1, BA.2, BA.3, BA4/5, and Delta were selected as targets. Probe–primer sets were designed to detect L452R, N501Y, T95I, delHV69-70, delLPP24-26, Ins214EPE, P681R, and E484A mutations. The developed reagent kit is a one-step multiplex assay containing three RT-PCR mixes including eight targets (target grouping per assay is shown in Figure 3).

Target selection and oligonucleotide selection were carried out, ensuring that no major polymorphisms interfered with the primers (the presence of polymorphisms needs to be constantly reviewed, given the rapid evolution of the virus). The final design of the multiplex is shown in Figure 3.

As a positive reaction control, a mixed control was created that includes eight analyzed targets. 

Samples confirmed to have SARS-CoV-2 RNA (as obtained using the AmpliSens^®^ COVID-19-FL reagent kit (Amplisens, CRIE, Moscow, Russia) were taken for analysis using a one-stage multiplex. Verification of target mutations was obtained using WGS and Sanger sequencing.

We examined over 100 samples for each target mutation to confirm the specificity and sensitivity for validating established methods in a single assay. Samples with a confirmed presence or absence of selected targets after sequencing (NGS and Sanger) were specifically selected for this study to confirm the results. Multiplex assays were tested after robust monoplex techniques were obtained. One-step PCR combined with the reverse transcription reaction was obtained after the optimization of the study conditions.

Table 1 shows the forward and reverse primers and probes used in this PCR methodology.

We used the information about the GC contents of the S-protein gene in this methodology and chose a locked nucleic acid (LNA) modification for the design of the probes. The Tm of the probes ranged from 65 to 70 °C. The IDT calculator was used for the evaluation of thermodynamic characteristics and secondary structures [26].

Our multiplex RT-PCR is adapted to Rotor Gene Q (QIAGEN, Hilden, Germany), DT-prime (DNA-technology, Protvino, Russia), and CFX96 Bio-Rad (Hercules, CA, USA) instruments.

### 2.6. Primer Synthesis for PCR Techniques

PCR primers were obtained by automated solid-phase synthesis via the phosphoramidite approach [27] on an ASM-800ET DNA synthesizer (Biosset LLC, Novosibirsk, Russia) according to custom protocols. Standard nucleoside phosphoramidites and solid-phase universal support were purchased from Glen Research (Sterling, VA, USA) and Am Chemicals LLC (Vista, CA, USA). Other reagents and solvents of reagent and USP grades were supplied by Sigma-Aldrich, Inc., headquarters: St. Louis, MO, USA), Acros Organics (Thermo Fisher Scientific, Geel, Belgium), Panreac Quimica SLU (Castellar del Vallès (Barcelona), Spain), and AppliChem GmbH (Darmstadt, Germany). All solvents were preliminarily prepurified before use by distillation and dried over molecular sieves.

After the standard procedure of ammonia cleavage/deprotection, oligonucleotides with the remaining DMT group were purified by solid-phase extraction on Glen-Pak cartridges (Glen Research, Sterling, VA, USA) and then processed for further application. The identity and purity of the sequences were assessed on a MALDI-TOF mass-spectrometer LaserToF LT2 Plus (Scientific Analysis Instruments, Stretford, UK) with a 3-hydroxypicolic acid substrate (Fluka, Charlotte, NC, USA).

### 2.7. Statistical Analysis

Microsoft Excel was used to calculate the statistical measures of the technique.

True Negative Results (Specificity) = True Negative/(True Negative + False Positive). True Positive Results (Sensitivity) = True Positive/(True Positive + False Negative). Accuracy = (True Positive + True Negative)/Total.

Total = True Negative +False Positive + True Positive + False Negative. 

The 95% confidence interval was calculated using the Clopper–Pearson exact method [28,29].

For low values, the Wilson’s criterion was used [28]. 

## 3. Results

### 3.1. Epidemiological Analysis of the Prevalence of Omicron Sublineages in Russia 

Based on cross-Russian data from VGARus, among the samples with a collection date in January 2022 the largest proportion (62.0%) accounted for samples identified as BA.1.1. However, the proportion of BA.1.1 had already decreased to 58.0% in February 2022, while BA.2 reached 19.0%. From March to June 2022, the structure of Omicron was dominated by the BA.2 sublineage, and its share during this period ranged from 53.0% in March to 74.0% in June 2022. In July 2022, there was a change in the dominant sublineage to BA.5.2, and its fraction was 63.0%, with BA.5.1 taking the second place in the structure at 10.0%. In August–October 2022, the distribution of the sublineages’ proportions did not change, as the fraction of the BA.5.2 variant only varied from 78.0% in October to 81.0% in September, and the fraction of the BA.5.1 variant only ranged from 6.0% in September to 7.0% in August.

Figure 4 presents the dynamics of the prevalence of SARS-CoV-2 Omicron variant sublineages in Russia from January 2021 to December 2022. The data were obtained from the VGARus database, which combines information about whole-genome sequences and sequences obtained using the Sanger method. The data were uploaded from all subjects in Russia. A territorial division by the organizations that carry out the sequencing and uploading of the data is also presented on the VGARus platform.

### 3.2. Results Obtained by Using One-Step Multiplex PCR

Table 2 presents the method for interpreting RT-PCR results. Some mutations have different incidences in different Omicron sublineages, and this should be taken into consideration in the application of the method. Data from outbreak.info [25] were used in the analysis of mutation frequency data.

This selection of target mutations allows not only for the differentiation of Omicron BA.1, BA.2, BA.3, and BA.4/BA.5 sublineages but also for the detection of new/atypical combinations of significant VOC mutations. Any atypical combination of mutations is a recommendation to perform sequencing of such a sample.

The PCR results are presented in Table 3. A total of 622 samples with different combinations of mutations were examined during the development of the methodology. The presence and absence of mutations were specifically confirmed by a sequencing method. Table 3 presents data for all formats (both single assay and multiplex assay and RT-multiplex assay), i.e., the calculation was performed exactly for the developed primer–probe combinations for each of the target mutations. Results not confirmed by Sanger or NGS sequencing were omitted from the calculation.

For Mix N501Y + T95I + L452R, the diagnostic specificity of the assay was 80.0–100.0% and the diagnostic sensitivity was 93.5–96.3%.

For Mix delHV69-70 + Ins214EPE + delLPP24-26, the diagnostic specificity of the assay was 88.7–100.0% and the diagnostic sensitivity was 97.8–100.0%. 

For Mix E484A+ P681R, the diagnostic specificity of the assay was 93.8–100% and the diagnostic sensitivity was 97.7–100.0% (the criteria for single-target P681R were confirmed experimentally in previous work [30]).

The interpretation of the results was carried out using Table 2.

Table 4 shows the results obtained from July to December 2022 using the RT-multiplex assay.

A total of 532 samples were analyzed using the developed technique from July to December 2022. The dominant role of the BA.4/5 sublineages was determined, the share of which in the studied samples was significantly higher than the shares of other sublineages. The results obtained make it possible to use the developed method as a screening tool for typing SARS-CoV-2 Omicron sublineages.

## 4. Discussion

The abundance of accumulated data on morbidity since the beginning of the COVID-19 pandemic helps to understand the peculiarities of the epidemic process among different populations and develop an optimal algorithm of anti-epidemic measures, taking into account the peculiarities of the spread of the pathogen on different territories. All this creates a basis for developing algorithms of anti-epidemic measures, establishing interaction between specialists from different fields of knowledge, and introducing molecular diagnostic methods and data analysis into the routine practice of epidemiologists. The general approach to the use of modern genomic and information technologies in the epidemiological surveillance of infectious disease agents makes it possible to extrapolate the results obtained to the sphere of the epidemiological surveillance of other infectious disease agents. 

Undoubtedly, whole-genome sequencing is by far the most informative and preferred method for pathogen surveillance [14] with subsequent storage of the results in the VGARus database and deposition in the GISAID. However, the use of less complex (e.g., Sanger sequencing) and cheaper methods (RT-PCR) in the surveillance of SARS-CoV-2 variants has not lost its relevance in some cases. Against the background of increasing morbidity, a large number of samples of biological material, or limited resources (different conditions for equipping regional laboratories), Sanger sequencing and RT-PCR screening methods can be an excellent tool for a rapid assessment of the epidemiological situation.

According to official statistics as of 31 October 2022, there were 21,429,506 cases of COVID-19 and a total incidence of 14,589 per 100,000 population in the Russian Federation [31].

The emergence of new SARS-CoV-2 variants in the territory coincided with an in-crease in population morbidity. 

The highest incidence rate occurred during the period of distribution of the Omicron variant in the territory of the Russian Federation; the incidence rate during this period was several times higher than in previous periods [8,32]. 

As of December 2022, more than 250 Omicron variant sublineages were identified among the samples of biological material obtained from patients from different regions of Russia in 2022 by whole-genome sequencing. 

This study does not include age, sex, and gender analyses, as these data do not influence the design of the PCR techniques.

The presented screening technique has a diagnostic specificity of 80.0–100.0% and a diagnostic sensitivity of 93.5–100.0%. It is a one-step multiplex assay for the detection of N501Y, L452R, T95I, delLPP24-26, delHV69-70, Ins214EPE, E484A, and P681R S-gene SARS-CoV-2 mutations. Depending on the combinations of the results obtained, there is a possibility to differentiate the BA.1*, BA.2*, BA.3*, and BA.4/5* Omicron sublineages and Delta variant SARS-CoV-2. 

The key mutations N501Y, E484A, and L452R are located in the RBD domain of the S-protein and are classified as VOC mutations by the WHO [33]. The N501Y mutation is known to affect the binding efficiency of the virus to the human cell ACE-2 receptor and increase the infectivity of the pathogen [34,35,36]. The N501Y mutation has been found in α, β, γ, and Mu lineages and is possibly advantageous from the virus evolution point of view. The N501Y mutation is also present in 90.0% of Omicron samples [20].

Mutations at the 452 (L452R/M/Q) and 484 (E484K/A) positions allow the virus to evade the immune response and avoid the action of monoclonal antibodies [37]. The E484K mutation has also been known for a long time; it was first described for the β variant and since then has regularly occurred in various variants: γ, Iota, Mu, etc. In the case of Omicron, another mutation, E484A, is represented at position 484, and it occurs in all sublineages (BA.1, BA.2, BA.3, BA.4, BA.5) and their “child” lineages [37].

The L452R mutation is also present in Delta, Kappa, and Epsilon variants (and L452Q in Lambda). Mutations in this position (L452R/M/Q) also occurred independently in several BA.2 sublineages. The L452R mutation is present in BA.4/BA.5 lineages. Since the summer of 2022, the Omicron lineage with a mutation in position 452 has remained dominant not only in Russia (Figure 4) but also in other countries [38,39,40,41]. 

Mutation P681R was chosen as a marker of the Delta lineage. The P681H mutation is characteristic of Omicron in this position. Presumably, mutations in this position increase the transmissibility of the virus [2,42].

The delHV69-70 deletion is characteristic of the Omicron variant and other VOCs (α, Eta), is located in the N-terminal domain of the S-protein, and also increases resistance to neutralizing antibodies and is associated with immunity evasion [43,44,45].

It should also be noted that any insertions and deletions are a convenient target for the development of a primer–probe system for RT-PCR detection. Such significant differences in the detected nucleotide sequences simplify the task.

Deletion in positions 24–26 was chosen as a BA.2 and BA.4/BA.5 lineages marker. This deletion, delLPP24-26, is located in the strategical antigenic site and presumably may help the virus to avoid the action of monoclonal antibodies or immune responses [46].

The insertion Ins214EPE is another convenient target for RT-PCR detection. Ins214EPE was not found in other lineages before the emergence of Omicron. This mutation is present only in the BA.1 lineage. It is assumed that the mutation may affect the T-cell response [47]. 

The T95I mutation was added to the PCR assay as a marker of the BA.1 and BA.3 lineages. It was also detected in the Delta sublineages (10.0–40.0%), as well as Mu and Iota variants. The impact of this mutation is currently unknown [48].

The technique was successfully applied to the analysis of more than 600 samples from whole area of Russia from July to December 2022. The data obtained are comparable with the results of whole-genome and fragment sequencing (Sanger) on other samples over the same period. The ability to adapt the methodology to new emerging options (e.g., BE*, BF*, BQ*, XBB*) is another advantage of the approach.

We have developed and validated a one-step multiplex RT-PCR method for screening SARS-CoV-2 samples, determining whether they belong to one of the Omicron sublineages, such as BA.1, BA.2, BA.3, BA.4/BA.5, and distinguishing these sublineages from the Delta line. This approach allows the testing of a large number of samples in a short time. It is planned to identify new emerging lines (XBB*, BQ*, etc.) for further development of the technique.

The quality management system of medical devices production at the CRIE is certified in accordance with the European standard EN ISO 13485 and GOST R ISO 13485 (National Standard), as well as in accordance with the Directive 98/79/EC of the European Parliament and Council on in vitro diagnostic medical devices. The development of the kits is based on WHO recommendations for the detection of SARS-CoV-2.

The surveillance system developed for SARS-CoV-2 (Figure 2) has been shown to be effective and reliable. The introduction of such algorithms into the practice of epidemiological surveillance of other infectious agents can be successful in some cases.

Single-mutation screening methods using PCR can also potentially be used to screen for genotyping other pathogens, identify drug resistance markers, etc.

PCR-based screening techniques naturally have their limitations. It is possible to get a negative result if there are polymorphisms in the primer or probe sites. It is for this reason that several loci should be added to the screening panel. If the variability of the pathogen is high, it is also necessary to review these sites constantly. In the case of SARS-CoV-2, differences between variants are no longer limited to the S-gene, so a genome-wide selection of mutation targets must be made, which is quite resource-intensive.

The method of typing pathogens using PCR has a number of limitations: when using it, the researcher receives information about the presence or absence of strictly defined mutations. This does not make it possible to mark the appearance of a new variant if the new mutations are located in other positions. Since, in this case, mutations are determined not by a direct method but by an indirect method, it is necessary to take into account the influence of various factors, such as the presence of inhibitors, fragmentation of the RNA contained in the sample, and others, on the efficiency of primers and probes. However, it should be noted that this method can be very effective for streaming samples in conditions of the unavailability of sequencing, time constraints on the study of materials, or a lot of samples, as was the case with an increase in the incidence against the backdrop of the spread of the Omicron variant.

The low cost, the flexibility, and the ability to adapt to new possibilities are the advantages of such methods. The rational use of a combination of all the techniques described gives a complete picture of the development of the epidemic situation in a large area.

## 5. Patents

1. VGARus database (certificate of state registration No. 2021621178 dated 2 June 2021);

2. VGARus (Virus Genome Aggregator of Russia) Service RuStrain (certificate of state registration No. 2021618856 dated 1 June 2021); 

3. The program for analysis and processing nucleotide sequence fragments to detect mutations in the SARS-CoV-2 genome (GEM:CoV-2) (certificate of state registration No. 2021617094 dated 6 May 2021);

4. Databank of the results of fragment sequencing of the SARS-CoV-2 genome (certificate of state registration No. 2021620892 dated 28 April 2021).

## Figures and Tables

**Figure 2 genes-14-01218-f002:**
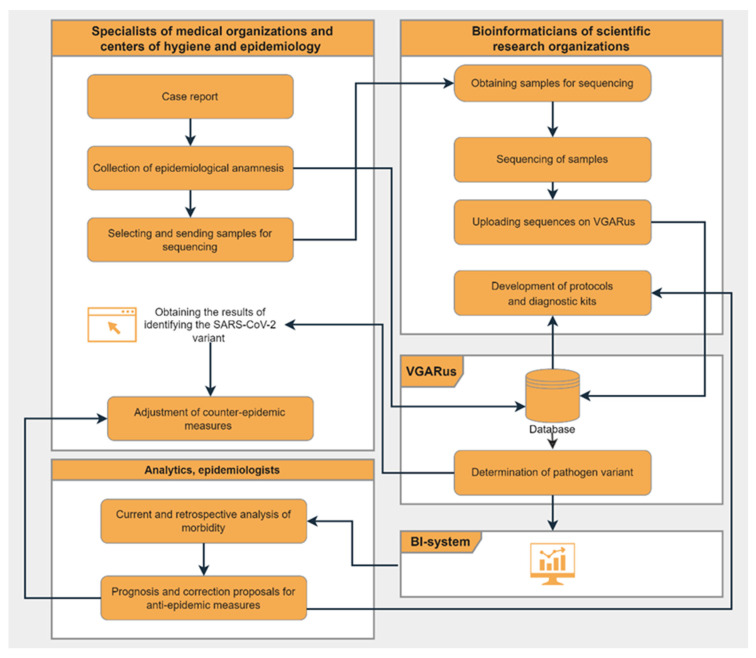
Summary scheme of VGARus integration into COVID-19 surveillance system in Russia (“Flowchart Maker & Online Diagram Software” [21] was used to make the figure).

**Figure 3 genes-14-01218-f003:**
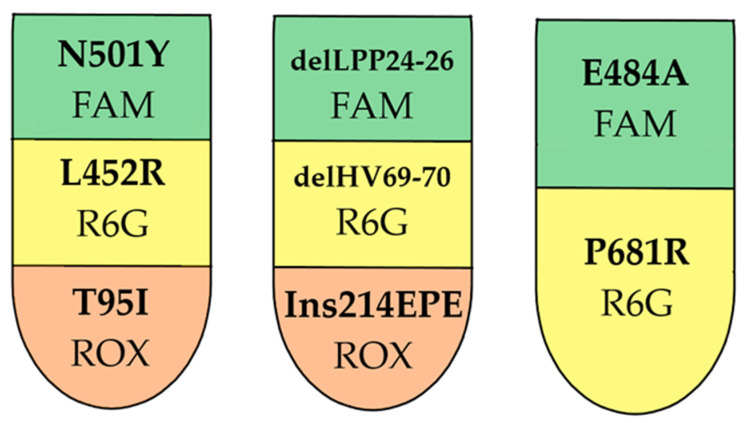
RT-PCR multiplex design. Presented targets and matching fluorophores.

**Figure 4 genes-14-01218-f004:**
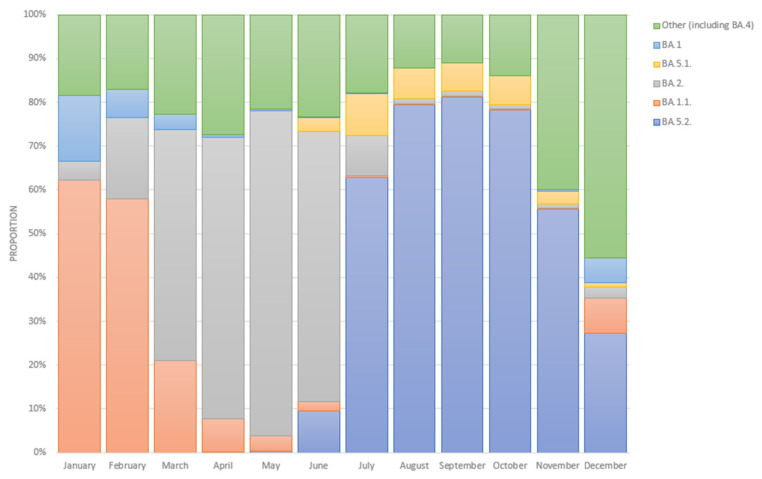
Month-wise breakdown of Omicron SARS-CoV-2 sublineages in Russia in 2022 (January–December, based on full-genome sequencing data) [18].

**Table 1 genes-14-01218-t001:** SARS-CoV-2 S-gene mutations and related primers and probes used in the multiplex RT-PCR.

SARS-CoV-2 S-Gene Mutations	5′-Sequence-3′
**L452R**	(F) GGC TGC GTT ATA GCT TGG AAT TCT
	(R) CCG GCC TGA TAG ATT TCA GTT GAA (P) * (R6G)AAT TAC CGG TAT AGA T (BHQ1)
**N501Y**	(F) CTG AAA TCT ATC AGG CCG GTA
	(R) GCT GGT GCA TGT AGA AGT TCA AAA G (P) * (FAM)CCC ACT TAT GGT G (BHQ1)
**T95I**	(F) GGT TTG ATA ACC CTG TCC TAC CA
	(R) GGG ACT GGG TCT TCG AAT CTA A (P) * (ROX)GCT TCC ATT GAG AA (BHQ-2)
**delHV69-70**	(F) GGA CTT GTT CTT ACC TTT CTT TTC CAA TG
	(R) TGG AAG CAA AAT AAA CAC CAT CAT TAA AT (P) * (R6G)TCC ATG CTA TCT CTG GGA C (BHQ1)
**delLPP24-26**	(F) CAC TAG TCT CTA GTC AGT GTG T
	(R) TGT CAG GGT AAT AAA CAC CAC G (P) * (FAM)GAA CTC AAT CAT ACA CT (BHQ-1)
**Ins214EPE**	(F) GGA CCT TGA AGG AAA ACA GGG TAA (R) CCA ATG GTT CTA AAG CCG AAA AAC C (P) * (ROX)TAG TGC GTG AGC CAG AA (BHQ2)
**P681R**	(F) GTA GGC AAT GAT GGA TTG ACT AGC TAC
	(R) TGC AGG TAT ATG CGC TAG TTA TCA GA (P) * (R6G)GCC GAC GAG AA (BHQ1)
**E484A**	(F) TTC AAC TGA AAT CTA TCA GGC CG
	(R) AGT TGC TGG TGC ATG TAG AAG TTC A (P) * (FAM)GTG TTG CAG GTG (BHQ-1)

* The probe sequence includes 4 to 7 LNAs.

**Table 2 genes-14-01218-t002:** Mutations selected for development of multiplex real-time RT-PCR.

	BA.1	BA.2	BA.3	BA.4/BA.5	Delta
**L452R**	-	-	-	**+**	**+**
**P681R**	-	-	-	-	**+**
**N501Y**	**+**	**+**	**+**	**+**	-
**delHV69-70**	**+**	-	**+**	**+**	-
**Ins214EPE**	**+**	-	-	-	-
**E484A**	**+**	**+**	**+**	**+**	-
**delLPP24-26**	-	**+**	-	**+**	-
**T95I**	**+**	-	**+**	-	- *

* This mutation was found in part of the SARS-CoV-2 Delta variant genomes.

**Table 3 genes-14-01218-t003:** Results of application of the developed technique on the archival samples and samples from July to December 2022.

	N501Y	L452R	T95I	Del LPP24-26	69-70	Ins214	E484A	P681R
**NGS/Sanger negative**	10	26	245	22	71	120	16	110
**True Negative (TN)**	8	26	241	20	63	120	14	110
**False Positive (FP)**	2	0	4	2	8	0	2	0
**Specificity/(CI)**	80.0% (49.0–94.4%) *	100% (87.1–100%) *	98.4% (95.9–99.6%)	90.9% (72.2–97.5%) *	88.7% (79.0–95.0%)	100% (97.0–100%)	87.5% (64.0–96.5%) *	100% (97.4–100%)
**NGS/Sanger positive**	213	192	46	235	276	3	255	0 **
**True Positive (TP)**	206	183	43	235	270	3	251	-
**False Negative (FN)**	7	9	3	0	6	0	4	-
**Sensitivity/(CI)**	96.7% (93.4–98.7%)	95.3% (91.3–97.8%)	93.5% (82.1–98.6%)	100% (98.4–100%)	97.8% (95.3–99.2%)	100% (43.9–100%) *	98.4% (96.0–99.6%)	-
**Total**	223	218	291	257	347	123	271	110
**Accuracy**	96.0% (92.5–98.1%)	95.9% (92.3–98.1%)	97.6% (95.16–99.06%)	99.26% (97.26–99.96%)	96% (93.36–97.8%)	100% (97.1–100%)	97.8% (95.2–99.2%)	100% (96.7–100%)

* Wilson score interval; ** For the P681R mutation, sensitivity and specificity were calculated earlier (during Delta variant circulation [30]). CI–confidence interval.

**Table 4 genes-14-01218-t004:** Results obtained from July to December 2022 using the SARS-CoV-2 screening multiplex RT-PCR typing method.

	BA.1	BA.2	BA.3	BA.4/5	Delta	Not Detected	Total	Confidence Level/Margin of Error *
**July**	2 (4.3%)	17 (37.0%)	0 (0.0%)	27 (58.7%)	0 (0.0%)	0 (0.0%)	46 (100.0%)	90%/10%
**August**	0 (0.0%)	2 (3.8%)	0 (0.0%)	51 (96.2%)	0 (0.0%)	0 (0.0%)	53 (100.0%)	90%/10%
**September**	0 (0.0%)	2 (1.4%)	0 (0.0%)	139 (98.6%)	0 (0.0%)	0 (0.0%)	141 (100.0%)	95%/2%
**October**	0 (0.0%)	0 (0.0%)	0 (0.0%)	181 (95.3%)	2 (1.1%)	7 (3.7%)	190 (100.0%)	99%/1%
**November**	0 (0.0%)	0 (0.0%)	0 (0.0%)	52 (100.0%)	0 (0.0%)	0 (0.0%)	52 (100.0%)	90%/10%
**December**	0 (0.0%)	3 (6.0%)	0 (0.0%)	47 (94.0%)	0 (0.0%)	0 (0.0%)	50 (100.0%)	90%/10%

* Quantitative representativeness of the selection.

## Data Availability

VGARus (https://genome.crie.ru, accessed on 31 May 2023), outbreak.info (https://outbreak.info/, accessed on 31 May 2023), and GISAID (https://www.gisaid.org, accessed on 31 May 2023).

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
