# Peer review of "Development and Application of Real-Time PCR-Based Screening for Identification of Omicron SARS-CoV-2 Variant Sublineages"

_genes, 2023, doi:10.3390/genes14061218_

Round 1

Reviewer 1 Report

An exciting study from the Akimkin group. The authors developed a multiplex assay for screening and detecting Omicron mutations. These mutations allow differentiation of Omicron sublineages such as BA.1 (delHV69-70, 19 T95I, Ins214EPE, E484A, N501Y), BA.2 (delLPP24-26, E484A, N501Y), BA.3 (delHV69-70, T95I, 20 E484A, N501Y), BA.4/BA.5 (delLPP24-26, delHV69-70, L452R, E484A, N501Y) and Delta variant 21 (L452R, P681R).

There are some issues:

1. Please explain if this system is available for other investigators.

2. How different is your system from others available on the market

3. Only Russian samples were tested. To avoid bias, can you include samples received from other countries? 

4. Authors used critical reagents available only in Russia. Can you validate data using kits from other companies? 

English is ok

Author Response

Thank you so much for your work! Please see the attachment

Reviewer 2 Report

The study entitled “Development and application of real-time PCR-based screening for identification of Omicron SARS-CoV-2 variant sub lineage” highlights the novelty of using molecular methods to track and analyze various sub lineages of the Omicron variant of SARS-CoV-2. The article presents the implementation of a comprehensive approach to obtain a complete epidemiological profile of the virus. Additionally, the development of a multiplex assay for screening and detecting specific mutations in the Omicron variant, allowing for differentiation between different sublineages and distinguishing them from the other variants, demonstrates an innovative contribution to the field of genetic analysis of COVID-19 variants. However, there are a few key areas where the authors can enhance their work. These points for improvement are as follows:

Abstract:

The abstract needs significant improvement in terms of its quality and tone. It should be written meticulously, encompassing the background, methods and materials, results, and novelty of the research. However, it is crucial to keep the discussion of these elements concise.

Introduction:

Overall, the manuscript provides a fine introduction, but there is a lack of clarity and coherence. The introduction jumps between different topics without providing a clear structure or flow of information. It begins with discussing the growing number of Omicron sublineages globally and in Russia, then shifts to describing the mutation profile of the Omicron variant, mentions the emergence of new sublineages and recombinant variants, and finally discusses the increased transmissibility and immune escape of the Omicron variant. The lack of a clear focus and organization makes it difficult for readers to follow the main points.

Give references in lines 61-64. There is no continuity between the two paragraphs starting from 61 to 75.

Give references in lines 83-85.

Write a few lines about your results in the last paragraph of the introduction.

The introduction briefly mentions the creation of a database for depositing nucleotide sequences and emphasizes epidemiology, but it does not clearly state the specific research focus or objectives of the study. It is unclear what new insights or contributions the article aims to provide.

Material and Methods

The text does not provide any information on the total number of samples used in the study.

While mentioning the certification of the quality management system is important, its inclusion in the material and methods section seems somewhat out of place. It might be more appropriate to mention this in the introduction or discussion section, where the relevance of quality management and adherence to standards can be better elaborated upon.

The mention of Illumina, Thermo Fisher Scientific, Oxford Nanopore Technologies, and MGI platforms for sequencing is very broad. It would be better to provide specific models/versions/chemistries used.

How did you check the validation of your in-house scripts?

In lines 134-135, it is written that the variants were either pushed into GEM or VGARus, what was the criteria or rationale for this distinction between sample processing.

A section mentions the selection of targets based on the prevalence of mutations in different Omicron sub lineages. However, it lacks specific information about how these targets were identified, the criteria used for their selection, and the sources of data used to determine their significance. The section mentions that the significance of mutations was considered, as well as the properties of the virus and the difficulty of detection implementation. However, it does not elaborate on these properties or provide a rationale for their importance in target selection.

Results

The comprehensive collection of data and the utilization of a multiomics approach have yielded significant insights. However, regrettably, the findings obtained from this extensive analysis have not been thoroughly deliberated upon.

There is no legend for table 3. Also In the footnote of Table 3, please include the information regarding the calculation of True positive/negative and False positive/negative numbers

Make a separate section of statistical methods used to analyze the results in material and methods.

Kindly ensure that the results and discussions are presented in a manner that highlights the main claims of the study, while also addressing any potential limitations. To effectively communicate the key aspects of a study to readers, authors should include a paragraph that outlines both the strengths and limitations of their research. By doing so, readers will gain a clear understanding of the scope and relevance of the study, as well as the potential areas for improvement or further exploration. This approach enables authors to present a balanced perspective and allows readers to make informed judgments about the significance and value of the research findings.

Author Response

(The authors gave the same response as above.)

Round 2

Reviewer 1 Report

All my comments are addressed.